# Overcoming Data Scarcity in Roadside Thermal Imagery: A New Dataset and Weakly Supervised Incremental Learning Framework

**DOI:** 10.3390/s25072340

**Published:** 2025-04-07

**Authors:** Arnd Pettirsch, Alvaro Garcia-Hernandez

**Affiliations:** Institute for Highway Engineering, RWTH Aachen University, 52062 Aachen, Germany; pettirsch@isac.rwth-aachen.de

**Keywords:** thermal imagery, traffic data collection, roadside cameras, weakly supervised learning, incremental learning, thermal image dataset

## Abstract

**Highlights:**

**Abstract:**

Roadside camera systems are commonly used for traffic data collection, yet conventional optical systems are limited by poor performance in varying weather and light conditions and are often restricted by data privacy regulations. Thermal imaging overcomes these issues, enabling reliable detection across all conditions without collecting personal data. However, its widespread use is hindered by the scarcity of diverse, annotated thermal training data, especially since fixed cameras installed at the side of the road produce very similar images with the same backgrounds. This paper presents two key innovations to address these challenges: a novel dataset of 11,400 annotated images and 142 unannotated video clips, the largest and most diverse available for thermal roadside imaging to date, and a weakly supervised incremental learning framework tailored for thermal roadside imagery. The dataset supports the development of self-supervised algorithms, and the learning framework allows efficient adaptation to new camera viewpoints and diverse environmental conditions without additional labelling. Together, these contributions enable cost-effective and reliable thermal-based traffic monitoring across varied locations, achieving an 8.9-point increase in mean average precision for previously unseen viewpoints.

## 1. Introduction

Reliable traffic data collection is essential for road planning, traffic simulation, safety analysis, and road asset management [1]. Among the various sources for collecting traffic data [2], camera systems have proven to be particularly advantageous when the trajectories of road users are of interest, for example, in safety analysis with surrogate safety measures (SSM) [3]. However, optical cameras are susceptible to weather conditions, rely on external lighting, and their use may be restricted by data protection regulations. In such situations, thermal imaging technology is advantageous as it is insensitive to weather and light conditions such as precipitation, darkness, sunlight, or shadows [4] and does not capture personal information.

There are already field solutions detecting road users in thermal images. Two well-known systems for urban real-time traffic detection are the FLIR TrafiSense2 [5] and the FLIR TrafiSense AI [6]. These systems can detect the presence and class of objects in predefined zones. This is sufficient for many applications, for example, for traffic light control. Applications like more detailed road-user-behavior studies call for flexible detection and tracking of objects to obtain the road users’ trajectory.

Systems enabling flexible traffic data collection beyond predefined areas rely fundamentally on effective and accurate object detection algorithms. In this context, object detection refers to the classification and concurrent pinpointing of objects within a video frame [7]. In the literature, there is a notable emphasis on applying these algorithms to traffic detection in RGB camera images, often concentrating on real-time detection [8] or sensor fusion [9]. In thermal imagery, many studies focus on applications such as monitoring drivers [10] or autonomous driving [11]. Researchers have addressed the challenging task of object detection in infrared images, which have fewer features than RGB images, through methods like combining RGB and infrared images [12] or enhancing established architectures by compressing channels and optimizing parameters [13]. While there is some research on detecting pedestrians [14] and parked cars [15] with roadside thermal cameras, there is still a significant gap in research on algorithms for that specific application.

Traffic detection with thermal images from roadside cameras poses an additional challenge. In contrast to applications with moving cameras (e.g., autonomous driving), the images in the datasets are very similar due to the fixed cameras, which poses a significant risk of overfitting, mainly because large datasets such as COCO [16], typically used for initial training, consist of images from RBG and, therefore, non-thermal cameras. Although Danaci et al. [17] have compiled 109 thermal imagery datasets, it appears that only the datasets from Balon et al. [18,19] and the AAU RainSnow Traffic Surveillance Dataset [20] are specifically available for this purpose. Balon et al. achieved promising results using YOLOv5 [21] and YOLOv7 [22] algorithms on their own dataset of infrared images from infrastructure cameras. However, both datasets contain only one location each and were recorded in half an hour and 45 min on one day [18,19], which is why it is to be expected that these models are overfitted to the viewing angles and environmental conditions used. In contrast, the AAU RainSnow Traffic Surveillance Dataset contains images from seven viewpoints [20]. However, due to the small number of about 2000 images, it is not expected to be sufficient to train generalizing models.

Data collection and annotation are costly, especially for infrastructure cameras, which require complex permits and potential road closures for installation. In RGB images, the scientific community is exploring self-learning approaches, often using larger teacher networks to generate pseudo-labels for training student networks [23], a method known as weakly supervised training [24]. These approaches often rely on image similarities [25] and sometimes on incorporating human knowledge [26]. Beyond new training, certain methods apply filtering [27] and tracking algorithms [28] to compensate for false-positive detections and misclassifications. Those algorithms rely on rules and constraints and may lack the generalization of retrained neural networks. Although works like Tang et al. [29] combine elements of both, there is limited research on solutions that fully leverage the unique properties of thermal roadside cameras, such as limited features and stationary cameras detecting primarily moving objects, within weakly supervised learning frameworks.

To develop a generalized algorithm for accurate traffic data collection across diverse locations, two key research areas require improvement in thermal image-based traffic detection. First, heterogeneous datasets are essential for effective viewpoint adjustment, necessitating distinct training and test data. Second, perspective adjustment methods are needed to address limited heterogeneous data, enabling universal application without additional human labeling

This paper enables reliable 2D object detection in thermal imagery from infrastructure cameras by addressing these two key gaps. First, it provides a unique dataset to address the lack of heterogeneous data in infrared images from varied traffic viewpoints. Second, it combines weakly supervised learning, incremental learning, and pseudo-label enhancement to a novel weakly supervised incremental training framework specially adapted to the challenges of roadside thermal imagery, allowing fast and robust adaptation to new camera positions and diverse traffic or environmental conditions.

## 2. Materials and Methods

### 2.1. Problem Description

A variety of models are trained on RGB images $X_{RGB}$ with high detection performances. $X_{RGB}$ consists of images $x^{RGB}$ and labels $y^{RGB}$ with y=(b,c), with b as the bounding box coordinates and c as the class label. In this work, these algorithms are adapted to the domain of roadside infrared images $X_{t}$. Within this domain, a comparatively small amount of human-labeled data $X_{t-A}$ exist. Transfer learning can achieve good results within $X_{t-A}$. However, sufficient heterogeneous test data are needed to evaluate how well these results generalize to different subdomains with varying camera locations, perspectives, and traffic situations $X_{t-B}$. This work addresses two problems: providing such testing data and adapting a model M trained with data from $X_{t-A}$ to $X_{t-B}$ without human annotation efforts.

### 2.2. Thermal Traffic Dataset

#### 2.2.1. Data Collection

Two thermal image camera models, the AXIS Q1942-E 10 mm [30] and AXIS Q1952-E 10 mm [31], were used to collect the data from the roadside. Both sensors record videos with 30 FPS and have 640 × 480 pixels resolution. The images are decoded in 8-bit format, where 255 is assigned to the hottest and 0 to the coldest pixel. These cameras were installed at 21 locations. These 21 locations include 12 inner-city spots, 5 locations on rural roads, and 4 on the highway. At all places, except two highway sites, the cameras were mounted 5–8 m high on street lamps or specially designed masts as shown in Figure 1. The streetlights or masts are located a few meters from the road. The other two highway cameras were mounted on road sign gantries at a similar height. The open-source tool LabelImg v1.8.0 [32] was used to create 2D bounding box annotations and class labels for every object of the classes: motorcyclist, car, bus, truck, pedestrian, bicyclist, or e-scooter.

#### 2.2.2. Data Splits

The dataset is split into three main parts $X_{t-A},\;X_{t-B1},\;X_{t-B2}$ (see Figure 2). Each part has unique camera locations and viewpoints. $X_{t-A}\;$ supports general domain adaption for models trained on RGB images. $X_{t-B1}$ is the target split with unseen camera locations and traffic. $X_{t-B2}$ is used to evaluate pseudo-label enhancement.

The first part $X_{t-A}$ includes 9000 images from 9 inner-city cameras. Approximately every 15th image is labeled, with at least a 0.5 s gap between frames, ensuring variations due to traffic flow. About 60% of the images of each camera are used for training and about 20% each for validation and testing. These cameras were positioned on frequently used streets, capturing various vehicles of all considered kinds of road users, such as cars, pedestrians, bicycles, motorcycles, trucks, buses, and e-scooters. This variety of object classes and their frequent appearance provide a solid base for learning generalized object representations. Given the high cost of annotation, 1000 images per camera, allocating about 600 images for training and roughly 200 for both validation and testing, were used. The second part $X_{t-B1}$ has 107 small non-annotated video clips from 8 camera positions. $X_{t-B1}$ also includes a split $X_{t-B1\_test}$, with 200 randomly selected images (0.5 s minimum interval) from additional videos of each camera ensuring to have the same amount of testing images per camera as in $X_{t-A}$. $\;X_{t-B2}$ contains 35 video clips from 4 camera positions, with approximately every 15th frame annotated, regardless of whether there is an object in the image, making it suitable for pseudo-label enhancement evaluation. $X_{t-B1}$ was chosen to contain as many different camera locations as possible with $X_{t-B2}$ kept smaller, as its purpose is to evaluate the pseudo-label quality. To cover all traffic situations, $X_{t-B2}$ contains one camera from inner-city and highway and, due to lower traffic flow, two from rural roads.

Table 1 illustrates the dataset’s composition, highlighting the heterogeneity of $X_{t-B1\_test}$. $X_{t-B1\_test}$ includes diverse viewing angles, times of day, and traffic situations, making it suitable for robust generalization evaluation. Figure 3 illustrates this variety, with on example image from each camera position $X_{t-B1\_test}$ covering all weather conditions.

#### 2.2.3. Data Evaluation

To determine if performance differences are due to testing dataset composition, $X_{t-A\_test}$ and $X_{t-B1\_test}$ are compared by class composition. Assuming larger objects are easier to detect, objects are additionally classified as hard (small), medium, and easy (larger). Objects under 128 pixels are hard, those 128–384 pixels are medium, and those 384 pixels or larger are easy. $X_{t-A\_train}$ and $X_{t-A\_val}$ splits are also analyzed to assess if the initial transfer learning dataset composition affects model generalization.

### 2.3. Training Framework

#### 2.3.1. Overview

The proposed training framework uniquely combines knowledge transfer between teacher and student networks, pseudo-label enhancement via motion-based filtering and temporal voting, and a remember module to prevent forgetting, all tailored to overcome the specific challenges of thermal roadside imagery. The framework combines soft-labels created by the teacher network and the remember module and hard-labels, which are not human-made but enhanced soft labels revised by the novel context module. As shown in Figure 4, the training method consists of four main parts: the teacher network, the student network, the context module, and the remember module (a frozen version of the student network). Following knowledge distillation [33], a large teacher model pre-trained on RGB images is fine-tuned on the annotated dataset portion. The teacher model is then fed unlabeled video clips from the $X_{t-B1}$ dataset. Detections on a subset of frames (every ~15th frame) are used as soft labels, leveraging inter-class relationships to train the student network [34]. The student is a smaller and more efficient version of the teacher network that learns from the soft-labels provided by the teacher network and the hard-labels provided by the context module. The student network is trained on new data but also benefits from the teacher’s prior knowledge, which helps to generalize better to new unseen environments. The student network also receives additional guidance from the remember module to avoid catastrophic forgetting. Teacher detections are also fed to the context module, which provides enhanced hard pseudo labels, as studies like [35] show their potential in weak supervision. The context module leverages unique properties of roadside cameras: moving objects, stable class across frames, and minimal traversal time. Using these heuristics, detections are filtered, and classifications adapted. This module is crucial for reducing false positives and improving classification accuracy.

The parts of the framework described above ensure high-quality pseudo-labels, allowing flexible, rapid adaptation to different sites. To prevent the network from forgetting previous knowledge and enable fast training without full retraining, a frozen version of the student (trained on prior locations), the remember module, is included in the framework. Each training batch also includes a small, random subset of images from prior locations, with the remember module providing soft labels. Previous context module labels are optionally used during validation. This combination of old and new data allows the model to retain valuable features while adapting to new environments.

#### 2.3.2. Basic Object Detection Architecture

Considering the practical demands of traffic analysis applications, which include real-time control of traffic management systems that require fast evaluation on edge devices or remote traffic studies that need battery-operated devices, an efficient architecture is essential. Thus, this work uses the YOLOv7-tiny model, which balances speed and accuracy, as the primary detection model [22]. Additionally, the YOLOv7-family contains widely recognized and validated models, ensuring the reproducibility and credibility of the presented results. For the teacher network, computational cost is less restrictive, allowing the use of the larger YOLOv7. Transfer learning, with a new detection head as suggested by the original authors of [22], was applied to adapt the pre-trained models to roadside thermal imagery. Nevertheless, the proposed method does not rely on any specific parts of YOLOv7 and can be adapted to other object detection architectures.

#### 2.3.3. Weakly Supervised and Incremental Learning Loss

##### General Loss

Since each batch includes images from both new and previous locations, the student network’s loss (L) term has two main components: $L_{Pseudo}$ which uses the output of the teacher model and context module to help the model to adapt to new locations, and $L_{Remember},$ which prevents the model from forgetting previously learned knowledge. A scaling factor α weights these two loss terms:(1)$$L=\alpha L_{Pseudo}+\left(1-\alpha \right)L_{Remember}$$

##### Pseudo-Label Loss

$L_{Pseudo}$ has three components, similar to the Yolov7 loss: objectness loss $L_{Obj}$, classification loss $L_{Cls}$, and bounding box regression term $L_{Reg}$. These parts are weighted by the factors β,γ, and δ as in the original paper. Since $L_{Reg}$ uses only boxes with true labels and the context module is expected to provide fewer false positives and more true positives, only those parts are used for regression. y is the network’s output and $y_{context}$ is the output of the context module. Objectness and classification losses include terms based on the teacher’s soft labels ${\tilde{y}}_{teacher}$ and the context module’s hard labels $y_{Context}$, with both weighted by objectness scores ($p_{obj_{i}})$. The classification loss balances context module outputs and teacher outputs using the teacher’s probability score ($p_{i})$, which multiplies objectness and classification scores based on the teachers input $x_{teacher}$. Since $y_{Context}$ and ${\tilde{y}}_{teacher}$ derive from teacher detections, each $y_{Context}$ detection has a matching ${\tilde{y}}_{teacher}$ detection. When only a soft label is available, the loss is weighted by the probability score, with all loss terms averaged. This yields the following for $L_{Pseudo},$ $L_{Obj\_Pseudo},L_{Cls\_Pseudo}$, and $L_{Reg\_Pseudo}$ [22]:(2)$$L_{Pseudo}=\beta L_{Obj\_Pseudo}+\gamma L_{Cls\_Pseudo}+{\delta L}_{Reg\_Pseudo}$$(3)$$L_{Reg\_Pseudo}={{mean(L}_{Reg}\left({y,y}_{context}\right))}_{i=0}^{i=m}$$(4)$$\begin{array}{cc} L_{Obj\_Pseudo} & ={mean({p_{obj\_i}(x_{teacher})L}_{Obj}\left({y,\tilde{y}}_{teacher})\right)}_{i=0}^{i=n}\\ & +{{mean((1-p_{obj_{i}}\left(x_{teacher}\right))L}_{Obj}\left({y,y}_{Context}\right))}_{i=0}^{i=n} \end{array}$$(5)$$\begin{array}{c} L_{Cls\_Pseudo}={mean({p_{i}(x_{teacher})L}_{Cls}\left({y,\tilde{y}}_{teacher})\right)}_{i=0}^{i=n}+{{mean(\varepsilon L}_{Cls}\left({y,y}_{Context}\right))}_{i=0}^{i=m}\;with\;\varepsilon \;=\left(1-p_{obj\_i}\left(x_{teacher}\right)\right)\;if\;matching{\;y\;}_{Context\;}exist\;and\;0\;otherwise \end{array}$$

##### Incremental Learning Loss

The $L_{Remember}$ equation uses the objectness, classification, and regression loss based on soft labels of the frozen student ${\tilde{y}}_{F-Student}$:(6)$$L_{Remember}=L_{Obj}\left({y,\tilde{y}}_{F-Student}\right)+L_{Cls}\left({y,\tilde{y}}_{F-Student}\right)+L_{Reg}\left({y,\tilde{y}}_{F-Student}\right)$$

##### Basic Loss Terms

Similar to Wang et al. [22], the objectness, classification, and regression loss terms for all losses were calculated as follows with $y_{i}$ either being hard or soft labels. $t_{Box}$ are the box coordinates passed by the context module. The objectness loss is balanced across different detection levels, as in the original work:(7)$$L_{Obj}=-[y_{i}log\left(p\right)+\left(1-y_{i}\right)log\left(1-p\right)]$$(8)$$L_{Cls}=-\sum_{i=1}^{C}y_{i}log\left(p_{i}\right)$$(9)$$L_{Reg}=\sum_{i=1}^{n}1-IoU(p_{Box},t_{Box})$$

#### 2.3.4. Context Module

The context module was specifically designed to address the unique challenges of thermal roadside imagery, such as low feature density and the stationary nature of cameras. By leveraging motion-based filtering and temporal class voting, it enhances pseudo-label quality in a novel and effective manner.

##### Sources of Error

The context module works to remove errors in teacher network detection. The three primary error sources are bounding box regression errors (producing misaligned boxes), background confusion, and class confusion (misclassified objects).

##### Improve Bounding Box Regression

Within the YOLOv7 model group, Wang et al. [22] found that YOLOv7′s bounding box regression is more accurate than YOLOv7-tiny’s, so YOLOv7 serves as the teacher model. Research exists on additional regression enhancement networks [36] and on methods that bypass bounding box regression in weakly supervised training [37]. In the proposed method, only an adaption to the different backgrounds is necessary. In the proposed method, only adaptation to different backgrounds is necessary since object classes, sizes, and aspect ratios remain unchanged. Thus, using a larger teacher network is effective and sufficient.

##### Remove Background Confusions

A combination of two methods removes false positive detections. First, non-moving objects are eliminated based on the assumption that false positives mainly occur in background areas. To achieve this, background images were generated using a temporal median filter (TMF) [38] based on (10), where k is half the total frame count, B is the background pixel value, and I is the image pixel value at position x,y. The teacher network is then applied to these background images. Detections that appear on the original image but not on the background image, using a matching threshold of 0.5 in IoU, are considered moving objects. In fixed-camera traffic monitoring, these moving detections are treated as true positives. There are limitations, however, when objects remain static due to traffic jams or parking. Such cases typically have high probability scores in the teacher network, reducing their impact on the overall loss calculation.

Second, non-moving object removal is combined with a minimum track-length filter that ensures each object is recorded for at least 30 frames. This works compares the proposed filter with box-density-based methods, like the IoU-based approach of Kim et al. [37] and the distance-based approach of Li et al. [27].(10)$$B\left(x,y\right)=median(\sum_{i=-k}^{k}I(x,y,i)$$

##### Remove Class Confusions

There are two key components in the approach to reduce class confusion: the tracking algorithm and the voting process. While tracking algorithms are an extensive research area on their own, this work primarily focuses on the voting problem. The SORT [39] algorithm is chosen for tracking, as it has shown strong results in real-time multiple object tracking and does not require additional, computationally costly models. The main assumption in using tracking algorithms for pseudo-label improvement is that an object maintains the same class throughout its track. Voting then determines the common class for these tracks.

This work evaluates three voting strategies: majority vote, maximum-score vote, and soft vote. Majority vote ignores probability scores and assigns the most frequently detected class (11), maximum-score selects the class with the highest overall score (12), while soft vote calculates the average score for each class across all detections in the track (13) [40].(11)$$x_{major}=arg\underset{c}{\max}\sum_{i=1}^{n}1(y_{i}=c)$$(12)$$x_{max}=\arg\underset{i\in \{1,..,n\}}{\max}p_{i}(c)$$(13)$$x_{Soft}=\arg\underset{c}{max}\frac{\sum_{i=1}^{n}p_{i}(c)}{n}$$

#### 2.3.5. Pseudolabel Creation

In contrast to actual training, the context module processes videos rather than individual images. Its output remains consistent throughout training. Thus, the pseudo-labels from the context module are pre-calculated. For this, a file with pseudo-labels (similar to the output of [32]) is created every 15 frames, starting from the 15th frame (α-frames). If this frame contains objects, it is saved as an image along with the label file; otherwise, the next α-frame is used. Notably, labels are only created after all objects detected in a specific α-frame have exited the tracking process. This ensures that the voting process includes all appearances of each object. Objects leave tracking if they have not been detected for 5 consecutive frames. For training, 600 images per camera were used, with 200 for validation. The images were selected as uniformly as possible (limited by the annotated frames) from each camera’s videos. The entire process is outlined in Algorithm 1.
**Algorithm 1.** Pseudo code for the creation of pseudo labels from the context module.1frame_num = 02annotations = {}3frame_objects = {}4# Iterate over all frames in the video5for curr_frame in video.all_frames():6
frame_num += 1  # Increment frame number7
# Process current frame8
curr_detections = detector(curr_frame)9
filtered_detections = detection_filter(curr_detections)10
tracked_objects = tracker.add_detections(filtered_detections)11
# Create pseudo-labels12
for curr_obj in tracked_objects:13

if curr_obj.last_detection() < frame_num—5: # Check if object is not detected for 5 frames14


for obj_frame_num in curr_obj.frame_numbers:15



if obj_frame_num % 15 == 0:  # Only consider frames divisible by 15 (alpha frames)16




if obj_frame_num not in annotations:17





annotations[obj_frame_num] = []18




annotations[obj_frame_num].append(curr_obj)19




frame_objects[obj_frame_num].remove(curr_obj)  # Remove from frame objects20
# Save labels if all objects have left the frame21
for frame_num_to_check in frame_objects.keys():22

if len(frame_objects[frame_num_to_check]) == 0:  # No remaining objects in the frame23


create_label_file(annotations[frame_num_to_check])24


save_image(images[frame_num_to_check])

### 2.4. Experimental Design

#### 2.4.1. Evaluation Metrics

##### General Evaluation Procedure

The training framework in this work comprises four modules: the teacher network, the student network, the context module, and the remember module. The two object detection networks are tested on the $X_{t-A}$ test set to assess adaptation to the thermal image domain and on the $X_{t-B1\_test}$ set to evaluate generalization, both before and after applying the training framework. The difference in performance before and after retraining the student network illustrates the proposed method’s benefit.

The context module enhances the teacher network’s output to produce high-quality pseudo-labels. Thus, the pseudo-label quality is evaluated per frame rather than evaluating filtering and tracking algorithms separately. To ensure that the context module optimization does not overfit specific camera positions and traffic situations, the $X_{t-B2}$ set was used for evaluation.

##### Bounding Box Regression

Bounding box regression is assessed using the Intersection-over-Union (IOU) metric. IOU measures the area of intersection divided by the area of union between the detected and ground-truth (GT) bounding boxes (14). This work uses the mean IOU (mIOU), calculated as the mean of all class-wise mean IOUs. For the other metrics, ground truth (GT) and detection are assigned based on an IOU threshold of 0.5 [41].(14)$$IOU=\frac{Area\;of\;Intersection}{Area\;of\;Union}$$

##### Classification

The mean Average Precision (mAP) is calculated as the mean of all average precisions across classes, where average precision represents the area under the curve in a precision–recall diagram. Precision (15) is the ratio of correct (true-positive) detections and all detections (true-positives and false-positives (FP)). Recall (16) is the ratio of true-positive detections (TP) and all ground-truth objects (true positives and false negatives (FN)). The precision–recall curve is derived using different recall levels (R) (17). At each recall level, only detections above a set score threshold (r) are included [16].(15)$$P=\frac{TP}{TP+FP}$$(16)$$R=\frac{TP}{TP+FN}$$(17)$$AP=\frac{1}{R}\sum_{r\in \left\{0,0.1,\ldots ,1\right\}}p_{interp}\left(r\right)$$

The mAP has the issue of score-threshold variation at the recall level. In practice, precision and recall at a specific threshold are prioritized. Therefore, the F1-Score (18), the harmonic mean of precision and recall, is used as an additional evaluation parameter. Similar to the AP and the IoU, the mean F1-score (mF1) is calculated. The score threshold is set at 0.25, as suggested by the authors of [22,42].(18)$$F_{1}=2\frac{P\cdot R}{P+R}=\frac{2TP}{2TP+FP+FN}$$

#### 2.4.2. Implementation Details

The code used in this work builds on the YoloV7 implementation by the original authors [22] and SORT tracking implementation [39]. Some algorithms for the training framework, pseudo-label generation in the context module, data processing, and evaluation were added to the base code. The code and hyperparameters for all experiments are available here: www.github.com/4rnd25/overcoming_data_scarcity_in_roadside_thermal_imagery.

Training and evaluation used an image size of 640 × 640, with resizing handled by the code from [22]. The hyperparameters, β, γ, and δ used in (2) are set identical to the original YOLOv7 paper [22], maintaining consistency with the original authors’ setup to ensure reproducibility and consistency with the established framework. Incremental training was tested with two strategies for α in equation (1): 0.5 for equal weights between the remember module and current training locations, and an adapted α, calculated as $\alpha =n_{Current\_locations}\;/\;(n_{Previous\_locaions}+n_{Current\_locations})$. For incremental training, the most recent version of the student network initialized the weights. The teacher network was not retrained; weights trained on $X_{t-A}$ were used for all training steps. For all soft labels (from the teacher network and the remember module) and for the input to the context module, a non-maximum suppression (NMS) process was applied on the teacher and frozen student outputs. The NMS was not class-agnostic and used a score threshold of 0.001 and an IOU threshold of 0.6. The context module only used detections above a 0.25 score threshold. All models were trained for 300 epochs, with the best models chosen based on the fitness score provided in [22] on the validation set. For incremental training, two strategies were tested for validation set selection: using the context module’s output on current training locations and combining current and all previous validation sets.

Training and inference times were measured on a server with an Nvidia Quadra RTX 5000 GPU and Intel Xeon E5-2640 v3 CPU (2.60 GHz, 16 cores). Additionally, inference time was assessed on an Nvidia Jetson Xavier NX Edge GPU in 15 W 6-core Power mode with Jetson Clocks activated.

## 3. Results and Discussion

### 3.1. Dataset

The labeled parts used for initial neural network transfer learning $X_{t-A},\;$ and the evaluation sets $X_{t-A\_test}$ and $X_{t-B1\_test}$ are emphasized in the dataset evaluation. Figure 5 displays the class distribution for these parts and the distribution of objects categorized as easy, medium, and hard to detect.

Class distributions are generally uneven across all analyzed dataset parts, with the car class significantly more represented than others. This distribution reflects traffic data collection applications, as cars are the most common vehicle type. Notably, the test set $X_{t-B1\_test}$, which includes heterogeneous viewpoints, contains fewer buses, pedestrians and cyclists but more motorcyclists and trucks. However, differences across difficulty levels are minor, especially in the most represented class, cars.

### 3.2. Performance Analysis

Figure 6 provides a summary of key results across experiments. The mAP on the $X_{t-B1\_test}$ dataset is shown, with crosses indicating each model’s performance at specific locations. Models were trained sequentially, location by location, in random order using data from $X_{t-B1}$ with the proposed framework, or collectively across all locations in $X_{t-B1}$. Training was conducted both without the remember module (red) and with it (blue and green). As described, two remember module options were tested: (1) α = 0.5 with validation data only from the current location or $X_{t-B1}$ (blue) and (2) an adapted α with validation data from all previous locations (green). Overall, performance on unseen data improved significantly. One observes that after adding five extra locations, the performance stabilizes and reaches similar results as when training on all locations. This supports the idea that more diverse locations lead to better detection quality and reduced overfitting. While Figure 6 reports the results during the process of incremental learning, Table 2 compares AP for the baseline model (Yv7t) trained on $X_{t-A}$, the incrementally trained model with adapted α and mixed validation data (Yv7t-PRIA), and the model trained on the entire $X_{t-B1}$ dataset at once using the adapted α and mixed validation set (Yv7t-PRA) after training on all locations. An increase of 8.9 percentage points in mAP was observed, with similar regression precision (78% mIoU to 79% mIoU) and a notably improved mF1-score (up by 7.5 percentage points). The analysis in Figure 7 indicates that this improvement is mainly due to fewer false positives in the background and better classification. While nearly all classes showed improvement, detection performance remains low for some classes. For instance, poor performance on the e-scooter class suggests that more class objects in the weakly supervised training data are needed and that the teacher network must initially detect all classes reliably.

Comparing the incrementally trained version with the one trained on the entire dataset at once, a slightly lower performance is observed. Additionally, the performance curve does not consistently rise, indicating that even with the remember module, some features are forgotten, especially since not every object class appears in all locations (e.g., no pedestrians in highway camera images).

Due to the high computational cost, mAPs are only reported for a single representative run per setup, consistent with common practice in object detection such as in [22]. Since the models were not trained from random initialization but from pretrained weights, remaining sources of randomness like the sample order are expected to have minor influence and would not explain the substantial improvement observed.

### 3.3. Comparison with State-of-The Art Datasets

Table 3 presents the results of the Yolov7-tiny model trained on the AAU Rain Snow Dataset (AAU) and the AutomotiveThermal Dataset (AT), evaluated on the test set of each specific dataset and the $X_{t-B1\_test}$ set. Bahnsen et al. [20] did not report specific results for object detection models on the thermal part of their dataset and did not define a test set. Therefore, 25% of each camera viewpoint was used for testing. In contrast, Baalon et al. [18] defined a test set and reported results with an mAP of 99.2, which was approximately reproduced here (99.7 mAP). Only the first published dataset by Balon et al. was considered since the second is no longer available online [18].

When applying the models trained on their original datasets to the $X_{t-B1\_test}$ set, a significant performance decrease was observed. The performance drop was particularly notable for the model trained on the AT dataset. This decrease reflects the homogeneous composition of Baalon et al.’s dataset, which resulted in poor detection quality on unknown data but exceptionally high performance on the original dataset. Such good performance indicates that both training and test sets within this dataset were too similar, leading to severe overfitting. This drop in performance provides a clear quantitative measure of overfitting, demonstrating how models trained on less-diverse locations fail to generalize to more varied data.

The AAU-trained model showed better performance on unseen data due to its more diverse dataset composition. However, the poor results on unknown data still suggest that the AAU dataset’s relatively small size of about 2000 images is insufficient for developing robust, generalizing algorithms.

### 3.4. Detailed Performance Analysis

#### 3.4.1. Performance of the Base Models

Comparing the performance of the larger YoloV7 model (Yv7) and the target model YoloV7-tiny (Yv7t) in Table 4, using the dataset $X_{t-A}$ for initial training on $X_{t-A\_test}$ and the more heterogeneous dataset $X_{t-B1\_test}$, leads to several conclusions. First, the significant drop in mAP and mF1 metrics highlights the generalization problem addressed by this work. Additionally, the use of the larger YoloV7 model as a teacher network is justified, as it performs better on unseen data (32.8 mAP vs. 31.4 mAP). Furthermore, the similar or better mIoU on $X_{t-B1\_test}$ reinforces that no additional bounding box regression improvements are necessary, as the teacher model’s bounding boxes with a high mIoU of 0.78 effectively guide the student.

Several factors could explain the lower performance on the more heterogeneous test dataset $X_{t-B1\_test}$, One reason may be the dataset composition itself, where class distribution influences performance. False positives have a higher impact on underrepresented classes, such as person, bicycle, and bus. For more represented classes, like motorbike and truck, one would expect an opposite effect, but the lack of this effect, combined with a generally similar distribution across difficulty levels (easy, medium, and hard), suggests other factors, such as insufficient training data. The high performance of the car class in $X_{t-B1\_test}$ also indicates that generalization may strongly depend on the quantity of training examples. These results further demonstrate the overfitting issue, as models trained on a more homogeneous dataset show significant performance drops when tested on more diverse and challenging data.

#### 3.4.2. Remember Module

##### Influence During Incremental Training

Figure 7 highlights the impact of the remember module in a general manner during the incremental learning process. Generalization performance is notably higher than that of the framework without this module across all training steps. Additionally, even on specific current training data, performance is superior for most locations. This improvement likely stems from learning more robust features due to training with diverse data (different viewing angles, objects, etc.) labeled by the remember module. Furthermore, the influence of human-annotated data enhances performance since $X_{t-A\_test}$ images are part of the training, and the loss function is structured to match previous outputs on human-labeled data.

##### Influence During Training on All Locations at Once

Table 5 compares the results of training on all locations in $X_{t-B1}$ at once, with and without the remember module, on $X_{t-B1\_test}$ and $X_{t-A\_test}$. Notably, similar performance is observed on $X_{t-B1\_test}$, but there is a significant difference on $X_{t-A\_test}$. This highlights the importance of the remember module, even with a larger, multi-location dataset, in achieving better performance on all data without compromising on new locations. Thus, performance even surpasses the baseline, which was trained only on $X_{t-A}$. data. However, overall performance on $X_{t-B1\_test}$ is lower than on $X_{t-A\_test}$, likely because strong supervision generally yields better results than weak supervision. Additionally, the diversity in weather, traffic, and backgrounds in $X_{t-B1}$ likely increases its complexity.

#### 3.4.3. Incremental Learning

The comparison of the blue and green data points in Figure 6 shows that both the scaling factor α and the validation dataset choice significantly affect incremental learning performance. Except for the first location, where only 1/8 of $X_{t-B1}$ locations were included in the validation set, the model with adapted α and mixed validation sets performed best.

Incremental learning without a remember module results in a sharp performance drop, likely because unlearned features are not relearned, and no loss term manages forgetting. The remember module with α = 0.5 helps mitigate this, but generalization performance still declines at some training steps. First, α = 0.5 may overweight new data, leading to overfitting. Second, generalization properties may be missing from the validation set, preventing the model from saving the best generalizing training state.

#### 3.4.4. Training and Inference Times

Table 6 shows the inference times with a batch size of 1 on Nvidia Jetson Xavier NX and the GPU server. On the server, model size has minimal impact on inference speed, particularly for efficient YOLO models, and YOLOv7 operates well above real-time speed (30 FPS). However, on the edge device, the difference is more substantial, with only YOLOv7-tiny achieving adequate speed (~25 FPS). On the edge device, the model was executed with onnx-runtime. Although further optimizations are possible, the general trend favors YOLOv7-tiny.

Examining training times, the teacher model and context module have little impact and, in some cases, even reduce training time. This is due to fast inference by the teacher model and prior labeling from the context module. Notably, only one measurement was taken, so shorter times may result from external factors, but the overall trend remains relevant. The remember module with both large and small validation datasets noticeably increases training time, likely due to an additional data loader for prior location images. The extended training time with larger validation sets indicates a bottleneck in data loading.

Incremental learning times per location are much faster on average, scaling with the volume of new data (images per new location, number of locations), which further demonstrates the advantages of this approach.

### 3.5. Evaluation of the Context Module

#### 3.5.1. Filtering Algorithms

The larger teacher model, together with tracking and filtering algorithms, was applied on the $X_{t-B2}$ dataset to analyze these algorithms’ ability to improve pseudo labels. Although $X_{t-B2}$ includes only four viewpoints, which raises overfitting risks, the dataset contains over 3000 objects, making the evaluations reasonably valid. The results do not rely on class-wise mean values of F1, precision, and recall when evaluating filter and voting algorithms, as these metrics should remain unaffected by class distribution in $X_{t-B2}$. For pseudo-labels, the focus is on overall counts rather than class-wise details.

All evaluations in Table 7 were conducted only on images with detections, as only these detections are used as pseudo labels for model retraining. This leads to a different object count across algorithms. The first observation is that neither the PDBE method based on [27] nor the SRRS method based on [37] worked effectively, especially when applying thresholds reported in [27,37] (PDBE thresh: 6, SRRS thresh 0.08). PDBE resulted in no filtering, while SRRS nearly filtered out all detections at these thresholds. A likely reason is thermal images’ generally lower feature density. Both methods rely on box density, which may not suit thermal images. Presumably, the lower feature density and pretraining on optical images cause certain areas in the image (e.g., some bright spots, even in the background) to register higher box density.

The TMF-based background filter and the minimum track length filter showed improved results, especially in precision, indicating that both algorithms significantly reduce false positive detections. However, both filters also reduce true positive detections, leading to a decrease in recall. As shown in Figure 8, true objects typically have a high probability score from the teacher network, so falsely filtered objects are unlikely to impact the student’s loss significantly. Since the background filter combined with the minimum track length filter achieves the highest F1 score, this combination is chosen as the filtering algorithm for this work.

#### 3.5.2. Voting Strategies for Tracking Objects

Comparing the different voting strategies (see Table 8), it is observed that soft vote and max score vote produce the worst results, likely due to a comparatively high number of false detections with high scores. Major vote proves more reliable as it includes all detections in the decision process. However, the minor change in the number of TPs indicates a low number of identity switches in the dataset $X_{t-B2}$, so the comparison is not entirely conclusive. Combined with the chosen filtering algorithms described above, the overall F1 score improves significantly by 6.4 percentage points.

### 3.6. Future Application

This methodology enables reliable, scalable traffic detection through thermal imaging in diverse environments, addressing the traditional limitations of location-specific training data and high retraining costs. By introducing a framework that quickly adapts to new viewpoints, this approach achieves high detection accuracy and broad generalizability with minimal human intervention.

The system provides dual advantages: quick adaptation to novel sites without the need for retraining from scratch and long-term performance enhancement through cumulative learning. Critically, it supports rapid updates for deployed systems, accommodating new locations or changing environmental conditions without compromising existing performance—a significant advancement for maintaining system reliability over time. Using video data collected under varied conditions, this framework demonstrates consistently improved detection across both new and previously seen viewpoints. Future implementations could further enhance generalizability by training on data from multiple new locations simultaneously, comparable to training on the whole $X_{t-B1}$ dataset. Additionally, the system’s ability to be deployed on edge devices such as the Jetson Xavier NX and powered by street lamps ensures low-power consumption, cost-effectiveness, and scalability, making it suitable for wide-scale urban deployment. This approach also holds promise for integration with smart city infrastructures, enabling real-time traffic monitoring and more efficient management of urban traffic systems.

Future applications could address the need for sensor failure detection, such as camera misalignment or thermal sensor issues, which are not automatically detected in the current framework. However, the architecture of the system is flexible, and these mechanisms can be incorporated in future versions.

## 4. Conclusions

The goal of this work is to advance robust object detection in thermal roadside imagery, aiming for reliable results across varied conditions. Our approach quickly adapts to new locations without human annotation and demonstrates significant benefits from this adaptability. The proposed training framework overcomes one of thermal imaging’s key limitations—the scarcity of publicly available training data—thus substantially lowering deployment efforts and enhancing reliability for traffic data collection. Key contributions include the following:

**Problem of Viewpoint Adaptation**: We demonstrated through performance comparisons, that diverse datasets are essential for achieving generalization and needed to identify the problem of missing generalization**Novel large Thermal Dataset**: We introduced a unique dataset with 9000 labeled images for transfer learning, 1600 for generalization evaluation, 800 for pseudo-label assessment, and 142 videos for weakly supervised learning—the most extensive thermal roadside dataset to date.**Innovative Training Framework**: We proposed a weakly supervised incremental learning framework that uniquely integrates knowledge transfer between teacher and student networks, pseudo-label enhancement through motion-based filtering and temporal voting, and a remember module to prevent forgetting, specifically designed to address the challenges of thermal roadside imagery.**Effective Pseudo label enhancement**: We conducted an in-depth analysis of false-positive filtering and compared voting strategies for pseudo-label enhancement tailored for roadside applications.**Incremental learning**: We verified the framework’s incremental learning benefit across eight different subsets of the prosed dataset.**Adaption to new viewpoint**: Overall, an improvement of 8.9 percentage points in the mAP on unseen data without any human labeling effort was achieved.

Future work will focus on refining the proposed method’s application in real-world settings, with particular attention to rare classes like e-scooter, Additionally making investigation on the optimal amount of images from new locations or situations needed. Furthermore, research on an automatic monitoring of the detection quality to recognize the need for retraining would enable practical application of a real-time traffic analysis system witch continuously improves without human interruption.

## Figures and Tables

**Figure 1 sensors-25-02340-f001:**
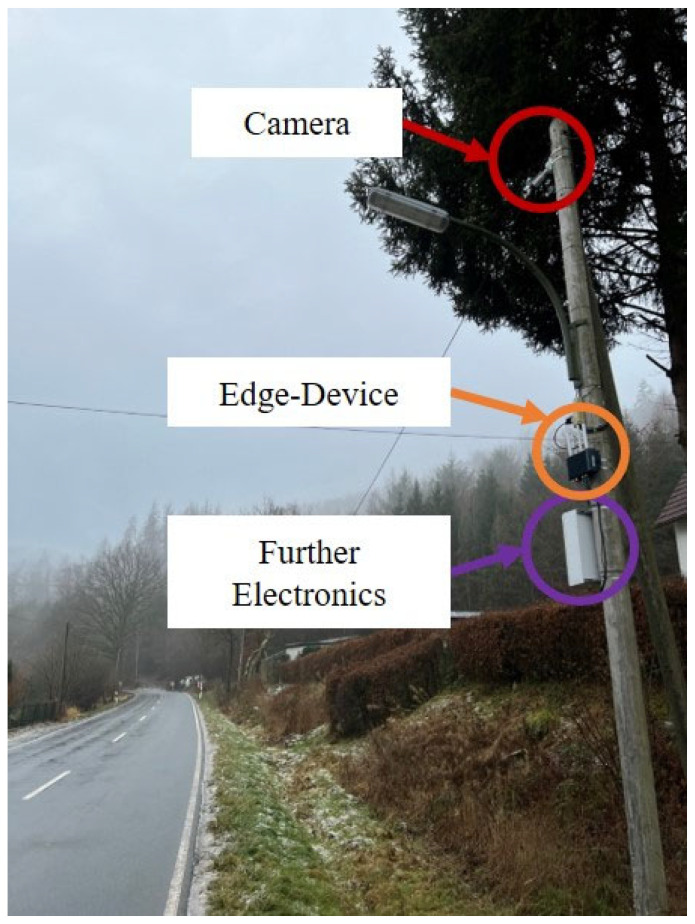
Photo of one of the thermal images cameras used in this study mounted at a rural road.

**Figure 2 sensors-25-02340-f002:**
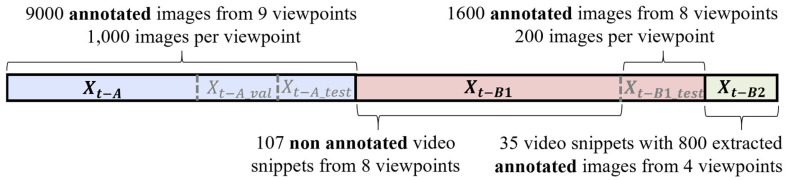
Dataset division: Each color represents a split. The first split is divided into training, validation, and test sets. The second split contains 107 non-annotated videos and 1600 annotated images from the same cameras at different times. The third split includes 35 videos and 800 annotated images extracted from them.

**Figure 3 sensors-25-02340-f003:**
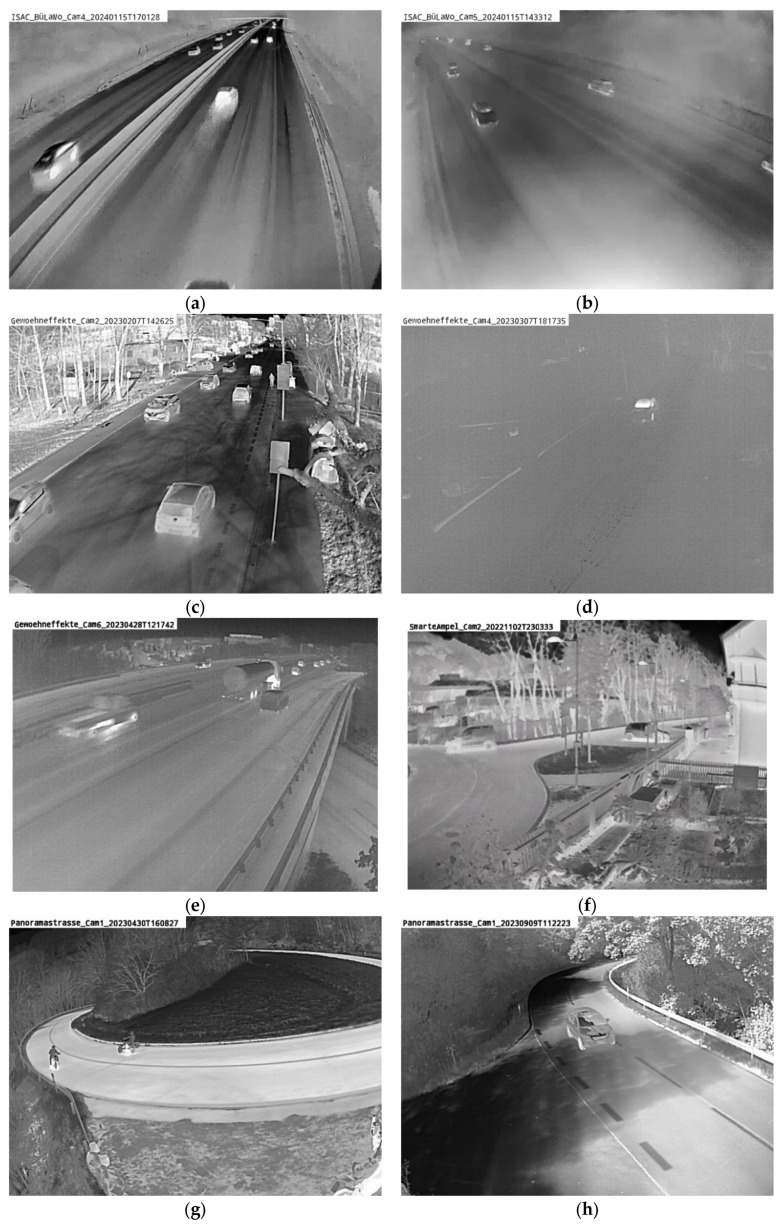
Different recording situations used in $X_{t-B1}$. (**a**) Rain/wet surface at highway. (**b**) snow at highway. (**c**) sun inner-city. (**d**) fog rural road. (**e**) wet surface highway. (**f**) sun inner-city. (**g**) sun rural road. (**h**) sun rural road.

**Figure 4 sensors-25-02340-f004:**
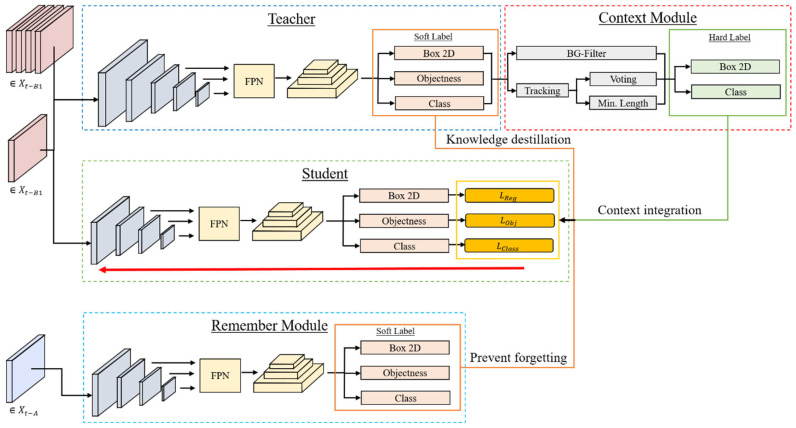
Overview of the proposed training framework. The teacher processes videos from $X_{t-B1}$, providing soft labels for key frames and input detections for the context module. The context module generates enhanced pseudo labels for the student network. The student trains on those key frames and additional soft labels from the remember module on previous data.

**Figure 5 sensors-25-02340-f005:**
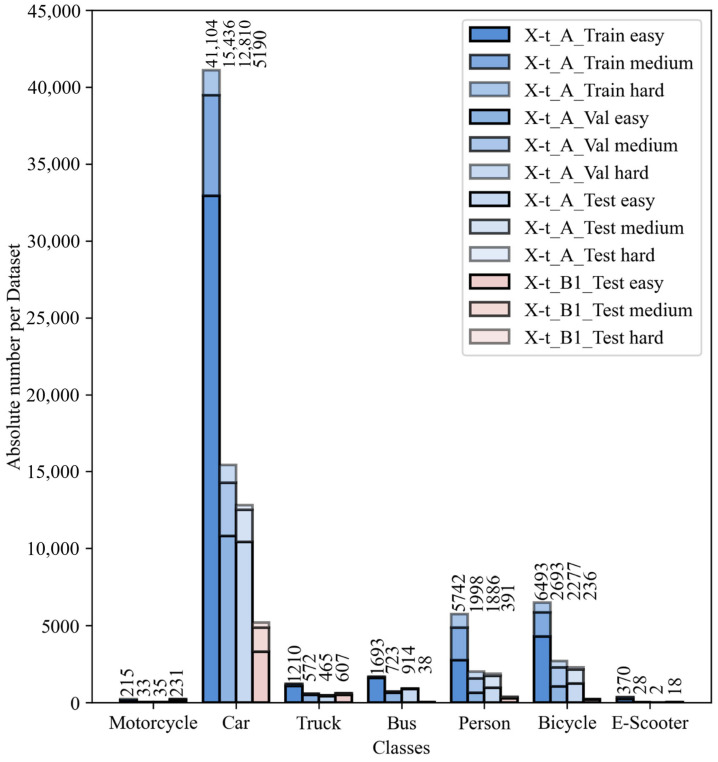
Distribution of human-labeled data sets $X_{t-A}$ for initial training and $X_{t-B1}$ for analysis of generalization abilities with respect to the classes encountered and the difficulty of recognizing the objects.

**Figure 6 sensors-25-02340-f006:**
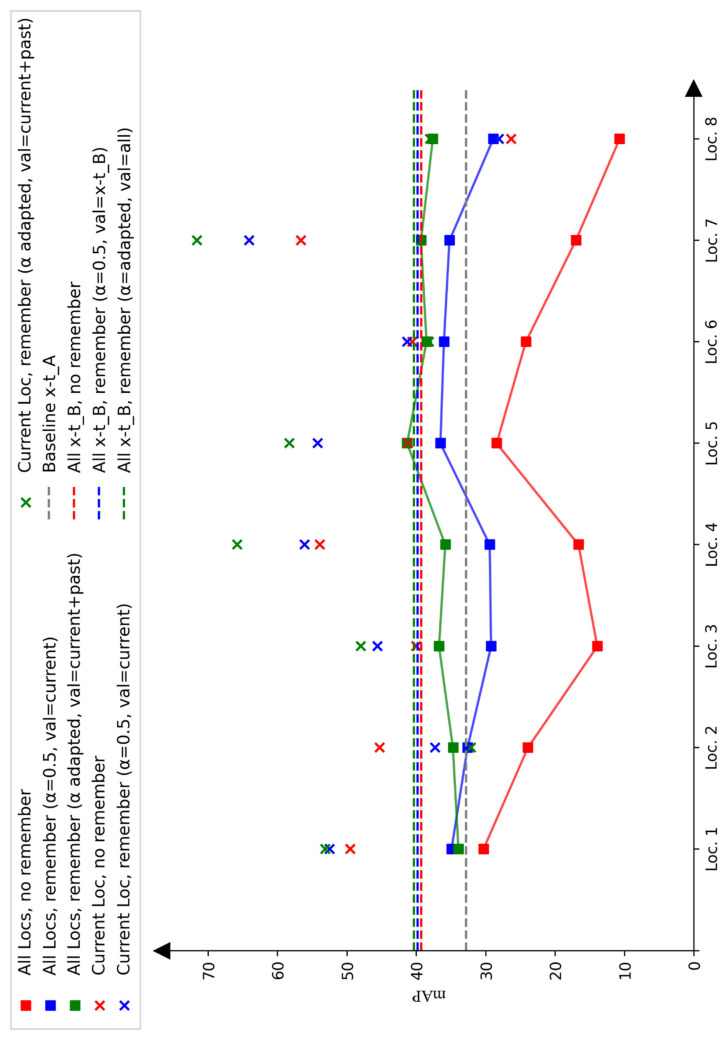
Performance of the difference model versions. The x-axis shows the process during incremental learning on the y-axis the mAP on the current data and the test set $X_{t-B1\_test}$ is shown.

**Figure 7 sensors-25-02340-f007:**
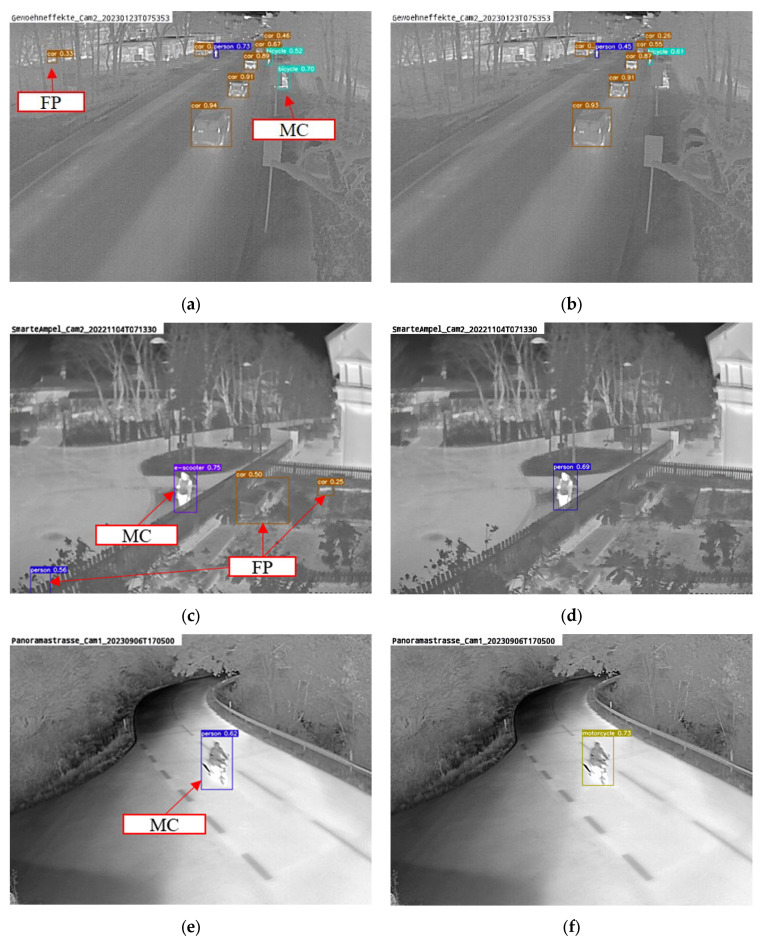
Example detections. (**a**) Yv7t with false positive (FP) and misclassification (MC). (**b**) Yv7t-PRA no FP and MC after retraining. (**c**) Yv7t with multiple FP and one MC. (**d**) Yv7t-PRA correct classification and no FP after retraining. (**e**) MC with Yv7t. (**f**) Correct classification with Yv7t-PRA.

**Figure 8 sensors-25-02340-f008:**
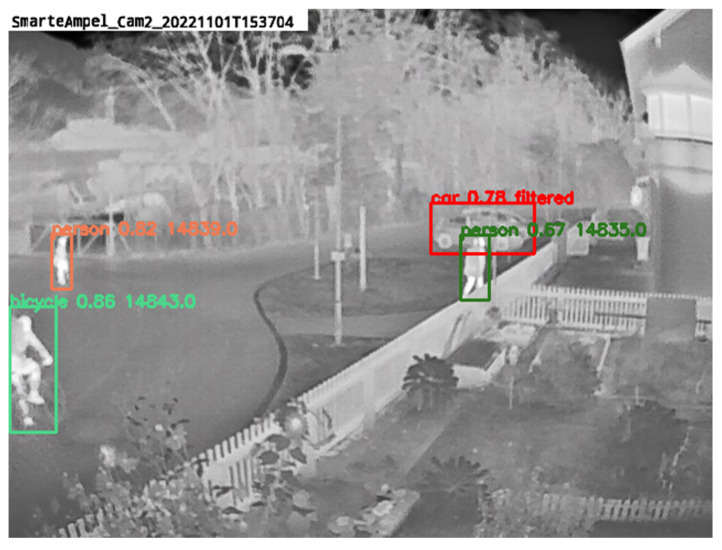
Example of the TMF filter combined with the minimum track length filter. The class name, the probability score, and an ascending counter number are shown above each object. Filtered objects are marked in red. One can see the filtered parked car, but it is also notable that the score is quite high with 0.78.

**Table 1 sensors-25-02340-t001:** Different recording situations, times of day, and weather conditions for the different dataset splits.

Split	Camera Positions			Day Time		Weather			
	Inner-City	Rural Road	Highway	Day	Night	Sun	Rain	Fog	Snow
$X_{t-A}$	9	0	0	✓	✓	✓	✓		✓
$X_{t-B1\_test}$	2	3	3	✓	✓	✓	✓	✓	✓
$X_{t-B1}$	2	3	3	✓	✓	✓	✓		✓
$X_{t-B2}$	1	2	1	✓	✓	✓	✓	✓	

**Table 2 sensors-25-02340-t002:** Results on the diverse test data $X_{t-B1\_test}$. Yv7t is the baseline tiny Yolov7 trained on $X_{t-A}$. Yv7t-PRA trained with pseudo loss and remember module as well as adapted alpha with mixed validation data and Yv7t-PRIA used the same set up but trained incrementally location by location. Results are given in percentage points.

Model	Data	Motorcycle	Car	Truck	Bus	Person	Bicycle	E-Scooter	Sum		
		AP	AP	AP	AP	AP	AP	AP	mAP	mF1	mIoU
Yv7t	t-B1	13.1	83.9	47.1	31.3	18.5	23.4	2.4	31.4	67.7	78.0
Yv7t-PRA	t-B1	33.2	87.1	63.5	35.0	26.8	34.9	1.7	40.3	75.2	79.0
Yv7t-PRIA	t-B1	22.1	87.2	59.9	38.8	24.5	28.3	2.6	37.6	74.5	78.0

**Table 3 sensors-25-02340-t003:** Results of the Yolov7-tiny model trained on the AAU rain snow dataset (AAU) [20] and the AutomotiveThermal dataset (AT) [18] on test sets of the original dataset and the $X_{t-B1}$ test set. F1, mF1, and mAP are given in percentage points.

Model	Data	Motorcycle	Car	Truck	Bus	Person	Bicycle	E-Scooter	Sum		
		AP	AP	AP	AP	AP	AP	AP	mAP	mF1	mIoU
Yv7t	t-A	78.1	88.9	66.2	65.7	42.3	52.2	0.2	56.2	73.2	78.0
Yv7t-AAU	AAU	/	77.5	36.9	4.2	0.2	32.3	/	30.2	67.1	0.76
Yv7t-AT	AT	/	99.7	99.8	99.7	99.8	/	/	99.7	97.6	0.84
Yvt7	t-B1	13.1	83.9	47.1	31.3	18.5	23.4	2.4	31.4	67.7	78.0
Yv7t-PRA	t-B1	33.2	87.1	63.5	35.0	26.8	34.9	1.7	40.3	75.2	79.0
Yv7t-AAU	t-B1	0.0	12.0	2.3	0.0	0.0	0.5	0.0	2.1	8.9	0.69
Yv7t-AT	t-B1	0.0	28.9	4.6	1.2	0.4	0.0	0.0	5.0	15.3	0.71

**Table 4 sensors-25-02340-t004:** Performance of the baseline teacher and student network on both main parts of the dataset. All results are given in percentage points.

Model	Data	Motorcycle	Car	Truck	Bus	Person	Bicycle	E-Scooter	Sum		
		AP	AP	AP	AP	AP	AP	AP	mAP	mF1	mIoU
Yv7	t-A	24.9	89.4	69.4	67.7	45.0	57.2	0.1	50.5	72.4	79.0
Yv7t	t-A	78.1	88.9	66.2	65.7	42.3	52.2	0.2	56.2	73.2	78.0
Yv7	t-B1	19.9	80.9	45.8	23.6	29.0	29.1	1.6	32.8	69.2	78.0
Yv7t	t-B1	13.1	83.9	47.1	31.3	18.5	23.4	2.4	31.4	67.7	78.0

**Table 5 sensors-25-02340-t005:** Comparison of the models trained on all locations of $X_{t-B1}$ at once using the remember module (Yv7t-PRA) and without the remember module (Yv7t-P).

Model	Data	Motorcycle	Car	Truck	Bus	Person	Bicycle	E-Scooter	Sum		
		AP	AP	AP	AP	AP	AP	AP	mAP	mF1	mIoU
Yv7t	t-A	78.1	88.9	66.2	65.7	42.3	52.2	0.2	56.2	73.2	78.0
Yv7t-P	t-A	54.1	84.1	54.5	57.8	23.7	25.9	0.0	42.9	68.7	75.0
Yv7t-PRA	t-A	77.3	89.4	64.1	65.4	48.5	58.4	0.1	57.6	75.6	77.0
Yvt7	t-B1	13.1	83.9	47.1	31.3	18.5	23.4	2.4	31.4	67.7	78.0
Yv7-P	t-B1	35.4	88.0	60.8	31.5	22.9	36.0	0.8	39.3	75.9	78.0
Yv7t-PRA	t-B1	33.2	87.1	63.5	35.0	26.8	34.9	1.7	40.3	75.2	79.0

**Table 6 sensors-25-02340-t006:** Inference and training times for the different model and training set-ups. Inference times were measured with batch size 1. Inference was only measured for baseline models since only training not the inference is effected by the proposed framework.

Model	Inference Time—Edge Device	Inference Time—Server	Training Time All	Mean Training Time Incremental
Yv7	0.2606 s	0.009 s	16.83 h	/
Yv7t	0.0393 s	0.0074 s	8.0 h	/
YvltP	/	/	7.35 h	1.44 h
YvltPR	/	/	13.02 h	2.36 h
YvltPRA	/	/	17.16 h	

**Table 7 sensors-25-02340-t007:** Results of the filtering algorithm. The table shows the mAP, F1, Precision, Recall, and the amount of TP, FP, and FN for the different filtering algorithms on the $X_{t-B2}$ dataset. F1, mAP, Precision, and Recall are given in percentage points.

Algorithm	F1	Precision	Recall	TP	FP	FN
YOLOv7	64.8	65.7	**63.9**	2064	1076	1168
Background filter	70.4	82.4	61.5	1942	415	1215
Min. track length filter	64.6	68.6	61.0	1935	887	1238
PDBE Thresh 6	64.8	65.7	**63.9**	2064	1076	1168
PDBE Thresh 100	62.6	64.6	60.7	1960	1073	1271
SRSS Thresh 0.08	5.5	30.0	3.0	3	7	96
SRSS Thresh 0.004	63.3	65.1	61.7	1989	1065	1237
Background filter min track length filter	**71.0**	**89.0**	59.1	1815	224	1256

**Table 8 sensors-25-02340-t008:** Results of the voting strategies. The table shows the mAP, the F1, Precision, Recall, and the amount of TP, FP, and FN when applying the different voting strategies on the $X_{t-B2}$ dataset.

Algorithm	F1	Precision	Recall	TP	FP	FN
YOLOv7	64.8	65.7	63.9	2064	1076	1168
YOLOv7 + Major Vote	**64.9**	**65.8**	**64.0**	2069	1074	1163
YOLOv7 + Max. score Vote	64.7	65.6	63.7	2059	1079	1173
YOLOv7 + Soft Vote	64.4	65.3	63.5	2052	1086	1180
Background filter w score thresh 0.5 + min track length filter + Major Vote	**71.3**	89.2	59.3	**1822**	221	**1249**

## Data Availability

The dataset is available under: https://doi.org/10.17632/66grzddyb2.1. The code can be found here: https://github.com/4rnd25/overcoming_data_scarcity_in_roadside_thermal_imagery.

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
