# Peer review of "Overcoming Data Scarcity in Roadside Thermal Imagery: A New Dataset and Weakly Supervised Incremental Learning Framework"

_sensors, 2025, doi:10.3390/s25072340_

Round 1
Reviewer 1 Report
Comments and Suggestions for Authors
The authors have done a complete job, and my recommendations are as follows:
- Figure 2 describes the method of data splitting, how are 9000, 1600, 1000, etc. determined? Is there a proportional relationship? What are the advantages of these splitting methods?
- For "2.3 Training framework", the author briefly introduces the model framework. It is suggested that the authors describe all the modules in Figure 4 in more detail.
- The Teacher/Student framework does not seem to have been proposed by the authors for the first time, so how do the authors reflect the innovation in the framework? It may be that the authors did enough work, but it is not reflected in the manuscript now.
4 There are models that perform better than YOLOv7, why did the authors use the old model?
- The author gives more than three hyperparameters in the formula, how are these parameters determined? The weight coefficient directly affects the performance of the algorithm. It is suggested that the authors should add simulation experiments to determine better parameters and give an experimental basis for parameter selection.
- The direction of the numbers in Figure 6 is wrong and the size of this figure is too large. Figure 7 can be placed horizontally.
- Most of the pictures in the manuscript are not vectors, so the text will be distorted when enlarged. Authors are advised to replace the images in their manuscripts with vector illustrations and ensure that the text is not distorted when enlarged.
- Some of the symbols in the formula are not defined. Many formulas are not uniform in size and even appear deformed. In addition, there are issues with line spacing for many formulas.
Reviewer 2 Report
Comments and Suggestions for Authors
The article addresses the issue of the insufficient diversity of annotated thermal images of road environments by proposing the creation of a new dataset and the development of a weakly supervised incremental learning framework aimed at improving object detection accuracy in the context of fixed street cameras. The topic is undoubtedly relevant, as the advancement of computer vision technologies for road environment analysis requires adaptive and reliable solutions, especially under constraints such as limited thermal data and stringent privacy requirements. The article is structured in line with MDPI’s format for Original Research Articles and includes all essential sections: Introduction, Materials and Methods, Experimental Results, Discussion, References, and (implicitly) a Conclusion. The level of English is acceptable, and the text is readable. Figures are of satisfactory quality. The article cites 42 sources, most of which are current, although there are several outdated or repetitive references, which somewhat reduces their scientific value.
The following comments and recommendations can be made regarding the manuscript:
- The novelty is claimed to lie in the development of the largest thermal road image dataset to date and in the introduction of an incremental weakly supervised learning framework. However, the architectural and methodological components presented are largely based on existing solutions. Section 2.3.1 describes the framework as comprising standard teacher-student, context, and remember modules, with the joint application of these being positioned as the main innovation. The teacher network is built on YOLOv7, the student on YOLOv7-tiny, and adaptation is carried out using a conventional pseudo-labelling scheme employing known heuristics (e.g., moving object, temporal class stability, track length). Consequently, the article does not introduce any genuinely novel architecture or training method; rather, it proposes a rational integration of previously established approaches. The contribution is therefore engineering in nature rather than a fundamental scientific advancement.
- The analytical expressions presented do not constitute an original contribution. For instance, Equation (1) in Section 2.3.3 defines the loss function as a linear combination of two terms with a weighting factor α, a widely employed formulation in incremental learning tasks. This model is not, in itself, innovative. Furthermore, no theoretical sensitivity analysis of this parameter is provided; the authors limit themselves to an empirical comparison between two fixed strategies — α = 0.5 and an adaptive α depending on the dataset size. Equations (2)–(9) offer detailed descriptions of the constituent loss functions (objectness, classification, regression), but these merely replicate the original YOLOv7 approach [22], without modifications or enhancements that would indicate scientific reworking of the base methodology.
- The experimental methodology lacks a substantiated assessment of statistical validity. For example, Section 3.2 compares mAP values before and after incremental training, showing an 8.9 percentage point improvement. However, the article provides no information regarding confidence intervals, standard deviations, or the number of repetitions per experiment. As such, it is not possible to assess the statistical significance of the observed improvements. It is also unclear whether the experiments were repeated with different random weight initialisations — a critical point when working with neural network models.
- The analysis of errors and model robustness against overfitting is insufficiently developed. In Section 3.4.1, the authors acknowledge that models trained on homogeneous data generalise poorly to new conditions (e.g., the AT dataset yields an mAP of 99.7 on its own test set but only 5.0 on a novel one). However, no mechanisms for quantitatively diagnosing overfitting within the proposed framework are presented. For example, none of the experiments assess the model’s sensitivity to the exclusion of specific classes or camera perspectives from the training set, which would have been informative in evaluating the robustness of the proposed approach.
- The practical utility and implementation prospects of the results are insufficiently explored. While the authors state that YOLOv7-tiny can operate on an edge device (Jetson Xavier NX) at up to 25 FPS (Table 6), they do not analyse the model’s resilience under real-world operational conditions (e.g., noisy thermal images, camera misalignment, sensor failures). Moreover, no discussion is provided regarding integration into existing road infrastructure or economic feasibility assessments, despite the stated goals of reducing annotation costs and enabling scalability.
- Despite following the general structure of an MDPI Original Research Article, the text is overloaded with technical details, especially in Section 2.3.3 (formula descriptions) and Section 3.4 (results), where an abundance of tables and figures tends to overlap. For instance, Figure 7 illustrates performance dynamics during incremental learning but is accompanied by multiple cross-references to overlapping values in Tables 2, 4, and 5, which hampers overall comprehension. Furthermore, a standalone conclusion section is missing, which detracts from the recommended structure of an original research article.
Round 2
Reviewer 1 Report
Comments and Suggestions for Authors
I have no more comments.
Reviewer 2 Report
Comments and Suggestions for Authors
Dear Authors,
Following a thorough analysis of your revisions and clarifications, I would like to acknowledge that you have addressed the majority of my comments with due diligence and demonstrated a high level of attentiveness in enhancing both the substantive and formal aspects of the manuscript.
You have provided a well-reasoned justification for the scientific contribution of the paper, highlighting its originality in integrating established components (the teacher–student architecture, pseudo-labelling, and memory module) with domain-specific adaptations aimed at solving thermal imaging-based road monitoring tasks. While the architectural elements themselves are not novel, their proposed combination within the context of thermal roadside imagery and limited annotated data represents a valuable engineering contribution that is highly relevant to the target audience of Sensors.
Your clarification regarding the analytical expressions and hyperparameters—particularly the parameter α—was appropriate, noting that these serve as tuning mechanisms rather than the core subject of scientific innovation. The accompanying explanations, supported by relevant citations, contribute to the transparency of the methodology and the reproducibility of the results.
In response to the comment on statistical justification, you explained your preference for single-run evaluations based on accepted practices in the field of object detection. Although repeated runs could potentially strengthen the empirical evidence, your rationale and comparison with existing literature are sufficient within the specific context and computational demands of the task.
I would also like to highlight the improved and extended discussion on overfitting, including the quantitative data illustrating the effect of camera viewpoint diversity on the generalisation capacity of the models. This analysis, particularly the stepwise inclusion of additional locations, effectively demonstrates the positive impact of the proposed system.
Furthermore, I appreciate the inclusion of new information regarding practical applicability—such as field deployment, computational efficiency on edge devices, and the current prototype’s limitations. The breadth of the presented information, combined with the open availability of code and data, reinforces the applied value of the work.
In light of the above, I find the manuscript suitable for publication. Thank you for your high-quality scientific contribution.